# Cracking Behavior and Deflections in Recycled-Aggregate Beams Reinforced with Waste Fibers Subjected to Long-Term Constant Loading

**DOI:** 10.3390/ma16103622

**Published:** 2023-05-09

**Authors:** Mateusz Zakrzewski, Jacek Domski

**Affiliations:** Faculty of Civil Engineering, Environmental and Geodetic Sciences, Koszalin University of Technology, Śniadeckich 2, 75-453 Koszalin, Poland

**Keywords:** waste fibers, waste aggregate, steel cord, fiber-reinforcement concrete, long-term study, concrete beams, deflection, crack

## Abstract

This report presents the results of long-term tests on concrete beams reinforced with steel cord. In this study, natural aggregate was wholly replaced with waste sand or with wastes from the production of ceramic products and ceramic hollow bricks. The amounts of individual fractions used were determined in accordance with guidelines for reference concrete. A total of eight mixtures were tested; these differed in terms of the type of waste aggregate used. Elements with various fiber-reinforcement ratios were made for each mixture. Steel fibers and waste fibers were used in amounts of 0.0%, 0.5%, and 1.0%. Compressive strength and modulus of elasticity were determined experimentally for each mixture. The main test was a four-point beam bending test. Beams with dimensions of 100 mm × 200 mm × 2900 mm were tested on a stand, which was specially prepared so that three beams could be tested simultaneously. Fiber-reinforcement ratios were 0.5% and 1.0%. Long-term studies were conducted for 1000 days. During the testing period, beam deflections and cracks were measured. The obtained results were compared with values calculated using several methods, considering the influence of dispersed reinforcement. The results enabled the best methods for calculating individual values for mixtures with different types of waste materials to be determined.

## 1. Introduction

Taking care of the natural environment is very important. Therefore, the environmental impact caused by the exploitation of mineral deposits should always be considered. Ideally, such exploitation should be limited as much as possible. This principle can be also applied in building works. To this end, a number of authors proposed the use of waste additives in concrete [1,2] or the replacement of traditional building materials with waste materials [3,4,5,6]. An example of such material is steel fiber (steel cord), which is a waste product of the recycling of car tires [7]. The effects of steel cord on many properties of concrete were already widely studied [7,8,9,10]. However, there were no studies on the effects of steel cord on the long-term properties of concrete. The long-term properties of concrete elements are very important when concrete is used for buildings, and they play an important role in the design of concrete structures. A good knowledge of the long-term properties of concrete allows structural engineers to more accurately predict the seriousness of serviceability limit states. Due to the amount of work involved, studies of the behavior of structures over time were performed less often than studies of short-term characteristics. This article presents the results obtained from such long-term testing. The influence of steel cord on the parameters of concrete beams were determined. In addition, three types of waste aggregate were used instead of natural mineral aggregate. As substitutes, waste from the production of porcelain and ceramic hollow bricks as well as waste sand were used. Waste sand was collected during mining of mineral aggregates by coarse extraction in a region of northern Poland [5]. Deflections and cracking processes of the beams were analyzed. Basic parameters of mixtures were determined. The compressive strength and modulus of elasticity of cylindrical samples were also determined. In this study, the authors focused on analytically determining the deflection of the beams over time. Several calculation methods were used to achieve this objective. The cracking of the beams over the course of the test period were also investigated. One of the aims of this study was to compare the results obtained using our testing methods with those obtained using selected existing methods to predict the behavior of beams over time.

## 2. Review of the Literature

The literature review concerned issues related to testing and determining the long-term properties of fiber-reinforced concrete. Improved methods for calculating the long-term parameters of fiber-reinforced concrete were long sought by scientific researchers. In 1995, Ezeldin and Shiah presented an analytical method for calculating deflection of SFRC beams over time [11]. They proposed the adoption of the stress–strain relationship identified in the research described in [12]. They tested beams with different contents of steel fibers, some with added silica. The obtained test results allowed for the determination of a formula describing the stress–strain relationship of the compressed fiber-reinforced concrete. The calculation algorithm is based on determining the beam curvature, taking long-term parameters into account. However, one of the assumptions of the method is that concrete deformations caused by shrinkage and creep are the same as in concrete without fibers. This method requires the determination of the tensile strength of fiber-reinforced concrete in tests, which is why the authors of this article decided not to use it. In 1994, Tan and Paramsivam proposed a modification of the method of calculating the long-term deflection contained in the ACI 318 standard [13]. The modification involves introducing an additional factor which takes into account the amount of fibers in the mixture. The values of this coefficient were determined on the basis of this research [14]. In 2005, Tam and Shah completed a 10-year study on beams with varying degrees of distributed reinforcement. The experimental results were compared with calculated results obtained using the method of effective modules which was modified by introducing a correction factor to take into account the aging of the concrete [15]. This method considers many factors affecting the deflection of beams over time, which is why the authors took it into account in their analyses. Bywalski and Kaminski proposed the following approach in 2011 [16,17]: based on Eurocode 2 [18], the relationships between steel fibers and moment of inertia were considered for both cracked and uncracked concrete. This method requires the use of computational programs due to the complex system of equations that must be solved. It is important to verify the calculations obtained by such methods using similarly sized elements similar to real structures. Due to the sheer physical requirements of this kind of research, such studies are rarely conducted, and are characterized by a small number of specimens [19,20,21]. 

Another important aspect of the behavior of concrete structures over time is the formation and propagation of cracks. In the case of fiber-reinforced concrete, scientists developed various methods for determining the width of cracks and their spacing. Some of these methods are based on Eurocode 2. They differ in the way the interaction between the reinforcing bars and the steel fibers is considered, and in how different boundary properties in cracked and non-cracked sections are assumed. One of these methods was first presented by Nemegeer at al. in 1995 [22]. They proposed a method for calculating the width of perpendicular cracks in beams with Dramix steel fibers. This method involved a slight modification of the proposal contained in Eurocode 2, i.e., for the calculation of stresses in the reinforcing steel, the tensile strength after cracking was assumed. The value of the equivalent tensile strength was determined from the Belgian standard NBN B15-238. The disadvantage of this method is that it is limited to fibers from only one manufacturer. The method proposed by Frosch is based on the same formula for determining the crack width as in Eurocode 2 [23]. The crack spacing is calculated as the product of the coefficient considering the situation, whether the minimum, average or maximum crack spacing is determined, and the appropriate distance of the center-of-gravity of the reinforcement in tension from the edge of the cross-section. Deformations of reinforcing steel are determined in terms of the ratios of stresses and moduli of elasticity. The use of this method in fiber-reinforced concrete elements is made possible only by appropriately determining the deformations of the reinforcing steel. In 2000, Vandewalle proposed another method for calculating the width of perpendicular cracks, based on Eurocode 2 [24]. The only difference is the adoption of an additional term in the formula for determining the average crack spacing. It is a simple method from a computational point of view.

Research on rheological properties is currently being carried out by many scientists. In addition to the calculation methods mentioned earlier, researchers analyzed the shrinkage and creep of concrete. These are two physical phenomena which largely determine the behavior of concrete elements over time. In 2019, Tošic et al. presented an experimental database and a revision of the fib model code method for determining creep [25]. In the same year, Geng et al. presented a revision of the method set out in Eurocode 2, based on tests of concrete with waste aggregate [26]. In 2015, an extensive analysis of computational methods of determining creep in concrete with waste aggregate was presented by Silva et al. [27] They concluded that the effect of waste aggregate on concrete creep had not been sufficiently studied for its effect on the deformation of concrete structures to be fully understood at that time. In 2020, Chen et al. presented a study of full-scale beams under high loads, on which both the creep coefficient and deflection of the beams were determined [28]. Finally, research into the long-term properties of concrete also involved analyses of increases in strength over time [29,30,31]. 

## 3. Materials and Methods

For research, waste from a plant producing ceramic hollow bricks (red ceramics); waste from a plant producing porcelain products (white ceramics) (Figure 1); and steel fibers obtained in the recycling process of end-of-life car tires (steel cord) were used. To obtain aggregates for reference concrete [32], ceramic waste was crushed on a jaw crusher and was then sieved using square-mesh sieves. The target aggregate content is presented in Table 1. During preparation of white ceramic waste, it was necessary to use a disc mill with a sieve with a mesh of 2 mm to obtain grains in the range of 0.125–2.0 mm. During the crushing and screening process, it was observed that obtaining sand fractions from ceramic waste was more labor-intensive. In subsequent mixtures, ceramic waste was replaced with local (Pomerania, North Poland) waste sand [33,34].

Portland cement CEM I 42,5 R was used in amounts of 400 kg/m³. Superplasticizer Silka ViscoCrete 5-600 was applied in amounts of 1% of cement weight. In mixtures made using porcelain waste, silica dust was used as an additive in amounts of 8% of the cement mass. Two types of fibers were used as dispersed reinforcement: hooked-end fibers of 50/0.8 mm steel; and steel cord. The properties of both these types of fibers were extensively studied [5,7,35,36]. Steel cord is produced during the recycling process of end-of-life car tires. This procedure involves mechanically processing the tires to produce rubber granulate, polyurethane–rubber composite, textiles, and steel cord. Steel cords differ in terms of the length and diameter of their fibers, due to the various types and sizes of tires (Figure 2). Steel cord specifications were determined in previous studies [7,9,10].

A total of 8 mixtures, which differed in terms of the aggregate used, were tested. For each mixture, samples were made without fibers and with fibers, in amounts of 0.5% and 1.0% by volume. For each mixture, cylindrical samples were made to determine the basic characteristics of the materials. The beams research program is presented in Table 2.

### 3.1. Testing Methods

The parameters of the tested mixtures—compressive strength and medium secant modulus of elasticity—were determined in accordance with standards [37,38]. These parameters were tested on cylindrical samples with dimensions of 150 mm × 300 mm. For each mixture, 6 samples were tested for compressive strength and 3 samples for modulus of elasticity. Deflections and cracks caused by constant long-term load were determined using beams with dimensions of 100 mm × 200 mm × 2900 mm, which were reinforced with two bars of RB 500 steel of 10mm diameter (Figure 3). The bars were supported approximately every 50 cm along the length of the beam; therefore, the random distribution of the cover [39] was not taken into account. The concrete covers of bars were of 15 mm thickness.

The cylindrical samples were demolded after 24 h and kept submerged until the test date. The samples were tested after 28 days. The beams were unmolded after 3–4 days. Subsequently, they were tightly closed with foil and regularly watered.

### 3.2. Long-Term Tests

The specimens were tested on a specially prepared stand (Figure 4). This stand was an original idea conceived by the authors of this article for the purposes of long-term study. Three beams could be tested simultaneously on a single stand, so the experiment did not take up much space. The beams were placed on top of each other. The middle beam was inverted. The use of articulated joints meant that individual beams were loaded by two concentrated forces only. Beams were subjected to a four-point bending test. Such a scheme enabled a quasi-constant bending moment to be obtained in the area between the forces. Two stands were placed on steel beams with a cross-section consisting of three welded IPE450 sections. The supports and load application points were made of solid steel shafts with a diameter of 50 mm. The lever consisted of two rods of 2500 mm length with a circular cross-section of 50 mm diameter. The load consisted of cast iron discs weighing 5 kg and 20 kg.

The long-term load was selected so that it exceeded the cracking moment on all beams. The load values are presented in Table 3. The beam with the greatest amount of dispersed reinforcement was placed on the bottom due to the fact that it carried the greatest load. Above this, a beam with dispersed reinforcement of 0.5% was positioned. Finally, a beam without dispersed reinforcement was placed on the top (Figure 5).

The study was divided into two stages. First, the instantaneous behavior of beams was analyzed; second, long-term deformations and cracking behavior were determined. In the first stage, the beams were loaded gradually, with increasing loads according to the scheme presented in Table 4. After each increase in load, measurements were taken using manual devices. Phases I, II, and III were completed in turn at 7-day intervals. Each beam had to be properly prepared before being placed on the stand. On the surface, measuring points were glued to measure deflections. After loading the beams to the selected level, instantaneous measurements were completed. Measurements were then taken at intervals to determine increases in deflections and deformations over time. The study of the behavior of the beams lasted for 1000 days.

Deflection was measured using dial gauges with an accuracy of 0.01 mm and calipers with an accuracy of 0.05 mm. Cracks and their course were monitored using a microscope with a 36-fold magnification and a resolution of 0.02 mm.

The stand was placed in the laboratory hall in which there were quasi-constant air conditions (Figure 6). Temperature and air humidity were measured hourly. The average temperature in the rooms where the stations were located was 21.6 °C with a standard deviation of 1.15 °C and air humidity was 40.7% with a standard deviation of 13.90%.

## 4. Calculation Methods

Predicting the behavior of concrete under load over time is difficult because it depends on many factors. In addition to the properties of the material itself, factors such as load level, air temperature, and humidity should all be considered. In this study, we compared measured deflection results with values calculated using the method contained in the Eurocode 2 (EC2) [18] standard, and also the methods for calculating the deflection of fiber-reinforced concrete elements proposed by Bywalski (Byw) [17] and Tan [15]. These latter two methods for the calculation of concrete deflections both consider the effect of fiber-addition on deflection. Bywalski’s method is a modification of the calculation method contained in Eurocode 2 and takes into account how the fibers increase the moment of inertia of the cracked section. Tan’s method considers, among other things, the effective modulus of elasticity of concrete calculated on the basis of the creep coefficient, which in turn is modified using correction factors which were obtained experimentally by Tan [15]. For calculations, a revised method which included a correction factor for calculations of the effective modulus of elasticity were used. The introduction of this factor was to take into account the effects of concrete aging. The behavior of cracks under long-term load was also analyzed. The actual distribution of cracks caused by long-term loading was compared with the results obtained by calculation methods. Predicted values for the spacing of cracks and their opening widths were calculated using the method contained in the EC2 standard [18], the Polish standard PN-B-03264 (PN-B) [40], and the method proposed by Vanderwalle (Van) [24]. This last method considers the correction due to distributed reinforcement.

## 5. Results

Table 5 presents values of the basic parameters—compressive strength, and medium secant modulus of elasticity—for the tested mixtures.

Table 6 shows crack widths in beams loaded to the assumed level after 1000 days. The number of observed cracks and their average spacing on each beam are also shown. The number of cracks and their spacing were analyzed only in the area between the forces, as shown in Figure 7. The measured values were compared with the values calculated using the EC2 [18], PN-B [40], and Vandewalle methods [24].

Figure 7 presents an exemplary illustration of the shape and distribution of cracks in beams placed on one stand.

The difference between the value of the crack opening and the crack spacing obtained during the tests and the calculation results are presented in Table 7. Table 7 and Table 8 present the values calculated according to the Formula (1):(1)$$\Delta =\frac{X_{test}-X_{calc}}{X_{test}}*100\%$$
where: $X_{test}$—value obtained during the research, $X_{calc}$—value obtained by calculation method.

Changes in deflection of all beams under long-term loading are presented as follows: Figure 8, Figure 9, Figure 10, Figure 11, Figure 12, Figure 13, Figure 14 and Figure 15 present comparisons of deflections in three beams placed on one stand, and comparisons of deflections of individual beams with values calculated using the method included in EC2 [18], the method presented by Bywalski [17], and the method of Tan [15]. Only long-term deflections are presented in the diagrams and in the table. Instantaneous deflection values are not included. The day on which the samples were loaded is treated as the beginning of the test, and the age of concrete at the time of loading was 28 days.

The graphs in Figure 8, Figure 9, Figure 10, Figure 11, Figure 12, Figure 13, Figure 14 and Figure 15 also contain the measurement errors. Error calculations were made which took into account the measurement uncertainties of the devices used and dimensional deviations resulting from the shape of beams.

Table 8 presents the values of beam deflections obtained after the beams were loaded and after 1000 days. A comparison of the deflection values obtained during the tests with the values calculated using the selected methods is also presented in Table 8.

## 6. Discussion

### 6.1. Cracks

The cracks observed during the tests were in line with expected values for both steel fibers and steel cord. With increases in the amount of dispersed reinforcement used, the number of cracks increased, the distances between them decreased, and their width also decreased. Significantly fewer cracks were observed in mixtures made using white ceramics, and their width was lower, not exceeding 0.08 mm. In all mixtures, steel cord gave similar results to fiber-reinforced steel fibers. A larger number of cracks was observed on samples made entirely of red ceramic aggregate, though their width was the same as in RCS mixtures and definitely greater than in mixtures with white ceramic aggregate. For RC materials, the best calculations of the width of cracks were obtained using the method included in PN-B [40]. For WC mixtures, better results were obtained using the EC2 method [18]. The Vandewalle method [24] overestimated the crack opening results; indeed, the calculated crack opening values obtained using the Vandewalle method differed by more than 50% from the actual results. However. the average values for crack spacing obtained by the Vandewalle method were close to the actual figures. In red ceramic aggregate-based mixtures, both the PN-B and Vandewalle methods failed to predict crack spacing and crack width. For the mixture with waste sand, it was difficult to choose the best method because all results varied, depending on the amount of fibers used.

### 6.2. Deflection

Depending on the waste material used, the deflection curves of the beams over time differed in terms of their shapes. In this regard, the mixtures could be divided into two groups. Mixtures based on red ceramic aggregate or on waste sand produced a more rounded curve in which deflection increased rapidly in the initial phase; the rate of increase in deflection then declined over time. Contrarily, mixtures based on white ceramic aggregate produced a more flattened curve in which the initial increase in deflection gain was lower and then gradually decreased over time. The shapes of these deflection curves indicated that, for many beams, the deflection had not yet stabilized. Further studies over longer time periods would, therefore, appear to be warranted.

### 6.3. Effect of Fibers on Long-Term Deflection

The effect of fiber reinforcement on beam deflection varied depending on the aggregate used. In mixtures based on red ceramic aggregate, the deflection was much greater, and the effect of adding fibers was more visible. This effect can best be highlighted by separately comparing the deflections of beams 2 and 3 on each stand. The bending moment on these beams was almost identical (B2: 5.24 kN·m; B3: 5.22 kN·m). However, in all mixtures based on red ceramic aggregate, the deflection of B2 beams was less than that of B3 beams. These values differed by 11.1%, 37.4%, and 19.4%, for RCf, RCc, and RCSc mixtures, respectively. For each of these mixtures, the deflection of the B3 beam—without fibers—was greatest, despite having the smallest bending moment. For the RSCs mixture, the deflections of the B1 and B2 beams were almost identical, differing only by 0.08 mm. For mixtures RCf and RCc, the differences between the deflections of B1 and B2 beams were −7.62% and −17.1%, respectively. For the mixture based on waste sand (Sc), the deflections of the B2 and B3 beams were identical (difference 0.03 mm). However, the deflection of the B3 beam was greater by 13.8%. This value was close to the difference between the values of bending moments acting on these beams, i.e., 13.0%. For this mixture, the addition of steel cord produced no significant effect on observed long-term deflection values. Similar results were obtained when steel cord was added to a mixture based on white ceramic aggregate and waste sand (WCSc). On this stand, the deflection of the B2 and B3 beams was also identical (difference 0.01 mm) and the deflection of the B3 beam was greater by 16.6%. Deflection values for all mixtures based on white aggregate only, and on white aggregate and waste sand, were very similar for all beams under long-term load. It should be noted that, in these cases, the effect of the fiber addition occurred only when fibers were used in amounts of 1.0% of the mixture volume. This is in line with the conclusions obtained in [9]. A tendency was observed that the deflection values of the mixtures with the addition of steel cord were lower than in the case of those mixtures to which steel fibers were added.

### 6.4. Calculating Methods

Good compliance was obtained by calculating the long-term deflection of beams made of white ceramic aggregate using Tan’s method [15]. The actual deflection after 1000 days differed from the calculated figure by a maximum of 10.88%. The greatest difference was noted for the WCf mixture. In the case of WCS, the differences were only 4.24% and 1.56%, for fiber reinforcement ratios of 0.5 and 1.0, respectively. In the case of the Sc mixture, high deflection compliance after 1000 days was obtained using Bywalski’s method [17], with differences between measured and calculated values of about 4%. From the point of view of construction safety, it is better to use Bywalski’s method. For mixtures based on white ceramic aggregate and waste sand, the safest option is to use the method contained in Eurocode 2. The method included in EC2 [18] gave results inflated by 10–60%. In the case of the RCSc mixture, EC2 overestimated results by 10.07%, 15.64%, and 29.45%, for B1, B2, and B3 beams, respectively. Finally, all the methods used underestimated the deflection of beams made of red ceramics.

### 6.5. Parameters of Mixtures

Both types of dispersed reinforcement resulted in a significant increase in compressive strength. In all mixtures, increases in the amount of dispersed reinforcement led to corresponding increases in strength, which were greatest in those mixtures in which waste sand was used. Among these mixtures, the greatest increase was recorded for the Sc mixture. In the others (RCS, WCS), the increase ranged from 9.61% to 15.59%. For mixtures made using only waste aggregate, the equivalent figure was about 3% for mixtures with a fiber-reinforcement ratio of 0.5, but greater than 6% for mixtures with a ratio of 1.0. Clear increases in the modulus of elasticity were noted for the Sc and RCSc mixtures. However, decreased values were recorded for the RCc and WCf mixtures. For the remaining mixtures, any changes in the modulus of elasticity were insignificant.

## 7. Conclusions

The results obtained from research and analyses lead to the following conclusions:-Aggregates produced from white ceramic waste and from waste sand can both be used to produce concrete with properties corresponding to ordinary concrete.-The addition of steel cord to the concrete improved its compressive strength. However, the influence of steel cord on the modulus of elasticity must be determined using a higher number of specimens.-For mixtures with aggregates based on waste sand and red ceramic waste (Sc, RC, RSC), crack width can be calculated using the method contained in the PN-B standard; however, the values so obtained were slightly underestimated in most cases. Considering the safety of construction projects, this calculation method should be used, with appropriate factors.-For mixtures with aggregate based on white ceramic waste (WC, WSC), crack width can be calculated using the EC2 method, which predicts actual values very well. The method from the PN-B standard also gives satisfactory results; in addition, it allows a certain safety margin for constructors.-Calculation of average crack spacing is best carried out using the method presented by Vandewalle, which produces figures most similar to actual values. However, these values were still overestimated. In the case of crack spacing, this means that this method should not be favored without appropriate corrections, because of structural-safety considerations.-For mixtures with aggregate from red ceramic waste, the prediction of deflection requires appropriate corrections to all calculation methods because the values obtained by calculation are definitely lower than actual figures.-The deflection of the beams with porcelain waste (WC, WSC) is best represented by the method presented by Tan. Using this method, calculated deflection values after 1000 days are very close to or slightly lower than the test results. With a slight modification of the method, it could be used to design optimized structures.

In conclusion, both waste sand and waste from the production of white ceramics give satisfactory results when used as aggregate for concrete. For red ceramic waste, further research and modification of calculation methods are necessary. The shape of the deflection curves indicates that, for many beams, at the end of the study period, deflection had not fully stabilized and values were likely to increase further. Further long-term studies are needed.

## Figures and Tables

**Figure 1 materials-16-03622-f001:**
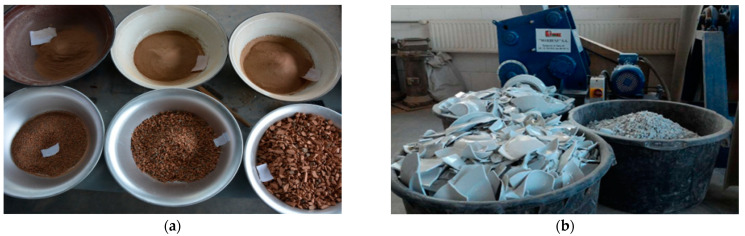
(**a**) Red ceramic waste after crushing and sieving. (**b**) White ceramic waste during crushing in jaw-crusher.

**Figure 2 materials-16-03622-f002:**
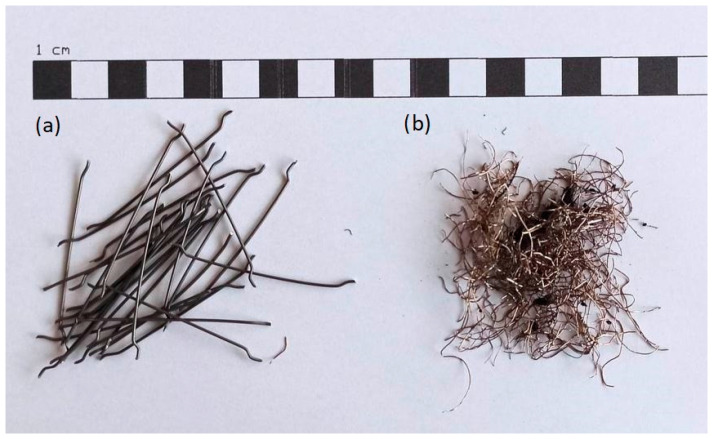
(**a**) Steel hook-end fibers, (**b**) Steel cord.

**Figure 3 materials-16-03622-f003:**
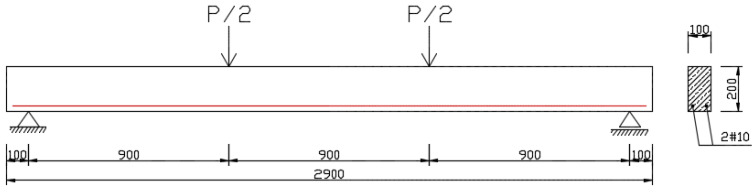
Geometry and static scheme of tested beams, details of conventional reinforcement.

**Figure 4 materials-16-03622-f004:**
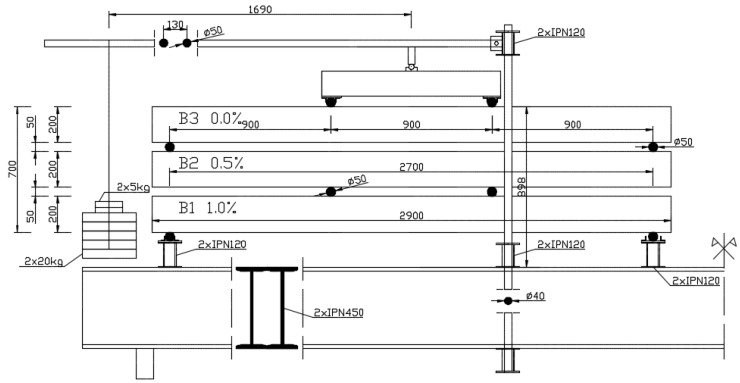
Set up for testing long-term deformations.

**Figure 5 materials-16-03622-f005:**
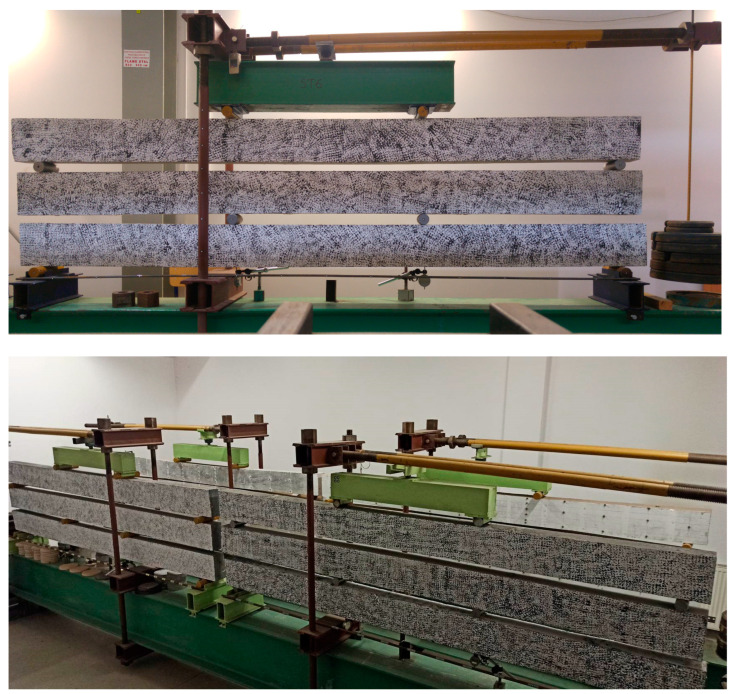
Stands for testing long-term deformations.

**Figure 6 materials-16-03622-f006:**
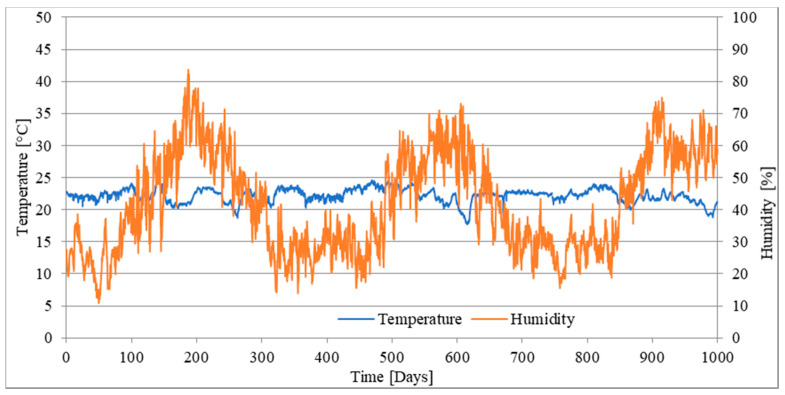
Air temperature and humidity during long-term test of mixture WCf.

**Figure 7 materials-16-03622-f007:**
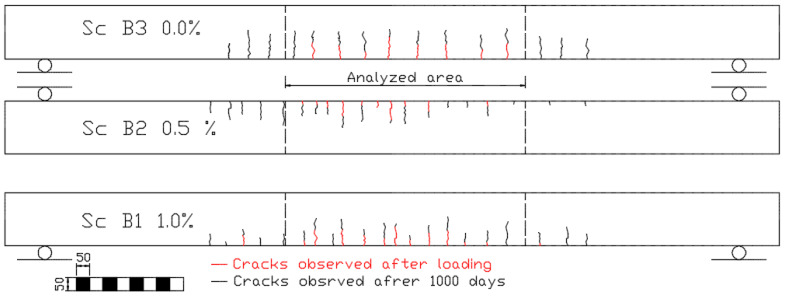
Example of crack patterns—mixture Sc.

**Figure 8 materials-16-03622-f008:**
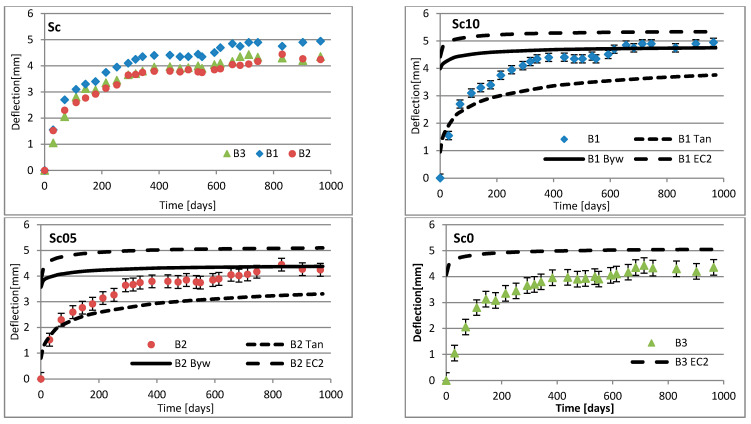
Diagrams of the deflection of beams in comparison to analytical methods—mixtures Sc.

**Figure 9 materials-16-03622-f009:**
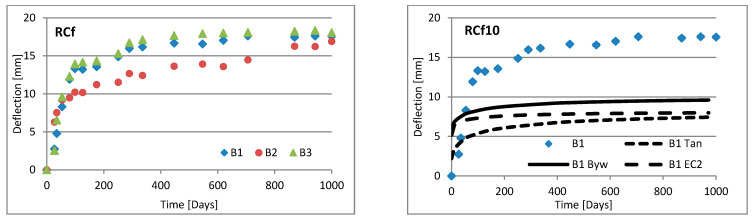
Diagrams of the deflection of beams in comparison to analytical methods—mixtures RCf.

**Figure 10 materials-16-03622-f010:**
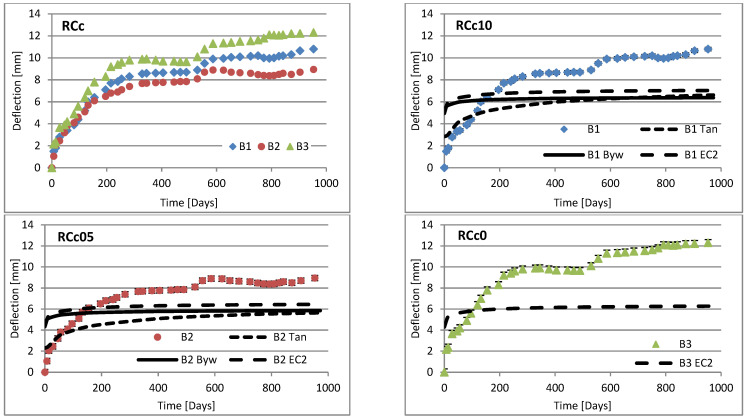
Diagrams of the deflection of beams in comparison to analytical methods—mixtures RCc.

**Figure 11 materials-16-03622-f011:**
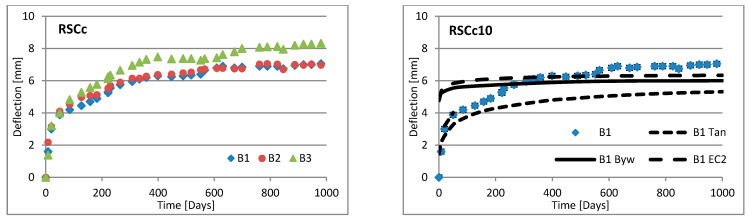
Diagrams of the deflection of beams in comparison to analytical methods—mixtures RCSc.

**Figure 12 materials-16-03622-f012:**
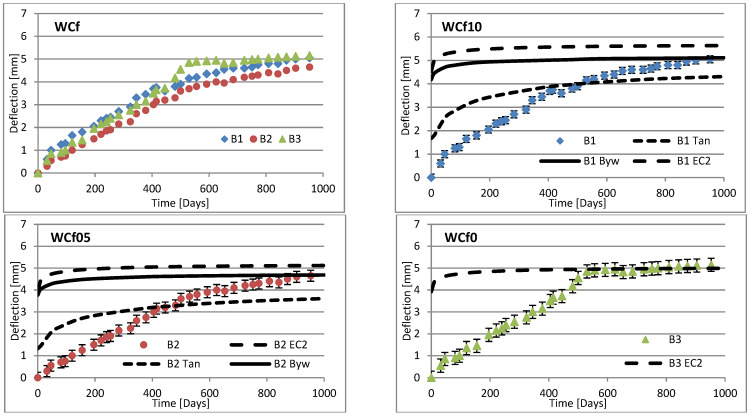
Diagrams of the deflection of beams in comparison to analytical methods—mixtures WCf.

**Figure 13 materials-16-03622-f013:**
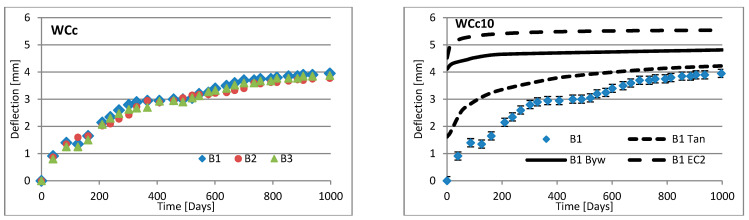
Diagrams of the deflection of beams in comparison to analytical methods—mixtures WCc.

**Figure 14 materials-16-03622-f014:**
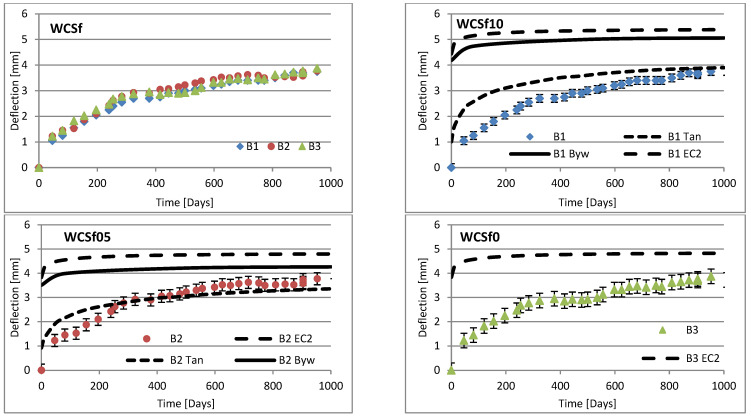
Diagrams of the deflection of beams in comparison to analytical methods—mixtures WCSf.

**Figure 15 materials-16-03622-f015:**
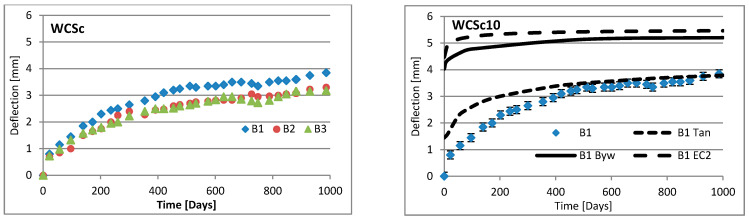
Diagrams of the deflection of beams in comparison to analytical methods—mixtures WCSc.

**Table 1 materials-16-03622-t001:** The amount of individual aggregate fractions in the mixtures.

Aggregate fractions range [mm]	0.125–0.25	0.25–0.50	0.50–1.00	1.00–2.00	2.00–4.00	4.00–8.00
Quantity [%]	10	15	12	15	20	28

**Table 2 materials-16-03622-t002:** Tested mixtures.

Symbol	Beam Number	Aggregate 0.125–2.00	Aggregate 2.00–8.00	Fiber-Reinforcement	Fiber-Reinforcement Ratio
Sc0	B3	Waste Sand	Natural aggregate	Steel cord	0.0%
Sc05	B2				0.5%
Sc10	B1				1.0%
RCf0	B3	Red ceramic waste	Red ceramic waste	Steel fibers 50/1.0	0.0%
RCf05	B2				0.5%
RCf10	B1				1.0%
RCc0	B3	Red ceramic waste	Red ceramic waste	Steel cord	0.0%
RCc05	B2				0.5%
RCc10	B1				1.0%
RCSc0	B3	Waste Sand	Red ceramic waste	Steel cord	0.0%
RCSc05	B2				0.5%
RCSc10	B1				1.0%
WCf0	B3	White ceramic waste	White ceramic waste	Steel fibers 50/1.0	0.0%
WCf05	B2				0.5%
WCf10	B1				1.0%
WCc0	B3	White ceramic waste	White ceramic waste	Steel cord	0.0%
WCc05	B2				0.5%
WCc10	B1				1.0%
WCSf0	B3	Waste Sand	White ceramic waste	Steel fibers 50/1.0	0.0%
WCSf05	B2				0.5%
WCSf10	B1				1.0%
WCSc0	B3	Waste Sand	White ceramic waste	Steel cord	0.0%
WCSc05	B2				0.5%
WCSc10	B1				1.0%

**Table 3 materials-16-03622-t003:** Characteristic of tested beams for each stand.

	Beams	Dimensions [mm]	Tensile Reinforcement [mm]	Fibre Reinforcement Ratio [%]	Load P [kN]	Bending Moment [kN·m]
1	B1	100 × 200 × 2900	2#10	1.0%	6.58	5.92
2	B2	100 × 200 × 2900	2#10	0.5%	5.82	5.24
3	B3	100 × 200 × 2900	2#10	0.0%	5.80	5.22

**Table 4 materials-16-03622-t004:** Load phases.

Phase	Load
I	B1 (own weight)
II	B1 + B2
III	B1 + B2 + B3
IV	B1 + B2 + B3 + Lever

**Table 5 materials-16-03622-t005:** Compressive strength and medium secant modulus of elasticity.

Mixture	Beam Number	Medium Secant Modulus of Elasticity [GPa]	Standard Deviation [GPa]	Compression Strength [MPa]	Standard Deviation [MPa]
Sc0	B3	37.89	1.84	42.03	1.88
Sc05	B2	39.50	0.54	60.51	0.72
Sc10	B1	42.20	1.20	72.53	8.76
RCf0	B3	16.56	0.32	37.74	1.55
RCf05	B2	16.97	0.11	38.82	0.69
RCf10	B1	15.04	0.31	40.24	0.92
RCc0	B3	20.05	1.12	47.55	2.75
RCc05	B2	18.74	1.83	49.40	1.91
RCc10	B1	18.65	0.12	50.52	2.12
RCSc0	B3	21.43	2.63	40.92	9.16
RCSc05	B2	23.30	0.62	48.09	0.95
RCSc10	B1	25.53	1.94	49.45	4.79
WCf0	B3	39.10	1.69	37.74	1.55
WCf05	B2	34.97	0.13	38.82	0.69
WCf10	B1	35.02	1.20	40.24	0.92
WCc0	B3	35.98	0.24	47.55	2.75
WCc05	B2	34.53	0.74	49.40	1.91
WCc10	B1	36.66	0.99	50.52	2.12
WCSf0	B3	41.15	0.72	59.20	6.04
WCSf05	B2	39.52	2.58	65.21	4.01
WCSf10	B1	40.09	0.63	66.72	4.56
WCSc0	B3	40.41	0.20	51.20	9.89
WCSc05	B2	40.91	0.22	56.12	8.23
WCSc10	B1	41.32	0.70	59.18	3.96

**Table 6 materials-16-03622-t006:** Cracks width and spacing.

Mixture	Beam Number	Crack Spacing [mm]				Max Crack Width [mm]					Number of Cracks
		Average	Average	Average	Maximum						
		Tests	PN-B	Van.	EC2	Instantaneous	After 1000 Days	EC2	PN-B	Van.	
Sc0	B3	93.8	76.18	69.25	95.50	0.09	0.12	0.074	0.097	0.054	10
Sc05	B2	72.9	77.64	70.58	97.99	0.09	0.14	0.069	0.092	0.049	14
Sc10	B1	64.1	78.28	71.16	99.08	0.05	0.08	0.082	0.108	0.059	14
RCf0	B3	46.6	70.00	63.69	85.11	0.09	0.12	0.063	0.090	0.047	19
RCf05	B2	47.7	70.32	63.93	85.54	0.09	0.12	0.063	0.090	0.047	19
RCf10	B1	48.8	69.68	63.35	84.46	0.10	0.10	0.073	0.103	0.055	19
RCc0	B3	44.3	72.48	65.89	89.22	0.08	0.10	0.064	0.090	0.047	22
RCc05	B2	58.1	72.17	65.61	88.68	0.09	0.10	0.062	0.089	0.046	15
RCc10	B1	49.6	78.60	65.66	88.79	0.10	0.12	0.075	0.112	0.055	21
RCSc0	B3	67.7	72.29	65.72	88.90	0.09	0.12	0.067	0.091	0.05	13
RCSc05	B2	52.3	73.50	66.76	90.85	0.06	0.12	0.066	0.091	0.049	19
RCSc10	B1	46.4	74.14	67.40	92.04	0.07	0.14	0.08	0.106	0.059	13
WCf0	B3	66.5	77.20	70.18	97.24	0.02	0.08	0.07	0.091	0.051	14
WCf05	B2	71.2	76.69	69.72	96.37	0.02	0.08	0.069	0.093	0.05	13
WCf10	B1	69.8	77.00	69.25	95.5	0.03	0.08	0.079	0.107	0.057	14
WCc0	B3	63.3	77.14	70.12	97.13	0.02	0.06	0.068	0.092	0.049	14
WCc05	B2	77.8	77.20	70.18	97.24	0.04	0.08	0.067	0.091	0.048	11
WCc10	B1	61.1	77.71	70.64	98.1	0.07	0.10	0.079	0.106	0.057	14
WCSf0	B3	76.9	77.83	70.76	98.32	0.08	0.10	0.069	0.092	0.05	12
WCSf05	B2	66.1	77.90	70.82	98.43	0.05	0.08	0.068	0.092	0.049	15
WCSf10	B1	57.6	78.02	70.93	98.64	0.03	0.04	0.081	0.105	0.058	14
WCSc0	B3	90.3	77.20	70.18	97.24	0.03	0.06	0.071	0.095	0.052	11
WCSc05	B2	71.8	77.52	70.47	97.78	0.04	0.08	0.07	0.093	0.05	11
WCSc10	B1	64.8	77.77	70.7	98.21	0.04	0.08	0.082	0.108	0.059	14

**Table 7 materials-16-03622-t007:** Comparison of calculation methods for cracks behavior.

Mixture	Beam Number	Δ Average Crack Spacing [%]		Δ Max Crack Width [%]		
		PN-B	Van.	EC2	PN-B	Van.
Sc0	B3	18.78	26.17	40.80	22.40	56.80
Sc05	B2	−6.50	3.18	50.71	34.29	65.00
Sc10	B1	−22.12	−11.01	−17.14	−54.29	15.71
RCf0	B3	−50.21	−36.67	47.50	25.00	60.83
RCf05	B2	−47.42	−34.03	47.50	25.00	60.83
RCf10	B1	−42.79	−29.82	27.00	−3.00	45.00
RCc0	B3	−63.61	−48.74	36.00	10.00	53.00
RCc05	B2	−24.22	−12.93	38.00	11.00	54.00
RCc10	B1	−58.47	−32.38	37.50	6.67	54.17
RCSc0	B3	−6.78	2.92	44.17	24.17	58.33
RCSc05	B2	−40.54	−27.65	45.00	24.17	59.17
RCSc10	B1	−59.78	−45.26	38.46	18.46	54.62
WCf0	B3	−16.09	−5.53	0.00	−30.00	27.14
WCf05	B2	−7.71	2.08	13.75	−16.25	37.50
WCf10	B1	−10.32	0.79	1.25	−33.75	28.75
WCc0	B3	−21.86	−10.77	−13.33	−53.33	18.33
WCc05	B2	0.77	9.79	16.25	−13.75	40.00
WCc10	B1	−27.18	−15.61	28.18	3.64	48.18
WCSf0	B3	−1.21	7.98	31.00	8.00	50.00
WCSf05	B2	−17.85	−7.14	2.86	−31.43	30.00
WCSf10	B1	−35.45	−23.14	−102.50	−162.50	−45.00
WCSc0	B3	14.51	22.28	−18.33	−58.33	13.33
WCSc05	B2	−7.97	1.85	12.50	−16.25	37.50
WCSc10	B1	−20.02	−9.10	−2.50	−35.00	26.25

**Table 8 materials-16-03622-t008:** Comparison of long-term deflection of beams obtained during research and with calculation methods.

Mixture	Beam Number	Test	Calculation Methods			Δ (Deflection) [%]		
		Deflection after 1000 Days [mm]	Deflection after 1000 Days [mm]			EC2	Tan	Byw
			EC2	Tan	Byw			
Sc0	B3	4.35	5.05	-	-	−16.09	-	-
Sc05	B2	4.24	5.10	3.31	4.38	−20.28	21.93	−3.20
Sc10	B1	4.95	5.34	3.76	4.75	−7.88	24.04	4.21
RCf0	B3	18.07	6.32	-	-	65.02	-	-
RCf05	B2	16.90	6.32	5.89	6.58	62.60	65.15	156.84
RCf10	B1	17.58	7.99	7.44	9.60	54.55	57.68	83.13
RCc0	B3	12.30	6.27	-	-	49.02	-	-
RCc05	B2	8.95	6.44	5.64	5.85	28.04	36.98	52.99
RCc10	B1	10.80	7.03	6.62	6.38	34.91	38.70	69.28
RCSc0	B3	8.32	5.87	-	-	29.45	-	-
RCSc05	B2	6.97	5.88	4.81	5.61	15.64	30.99	24.24
RCSc10	B1	7.05	6.34	5.32	6.00	10.07	24.54	17.50
WCf0	B3	5.15	4.99	-	-	3.11	-	-
WCf05	B2	4.65	5.12	3.61	4.69	−10.11	22.37	−0.85
WCf10	B1	5.05	5.63	4.31	5.11	−11.49	14.65	−1.17
WCc0	B3	3.87	5.38	-	-	−39.02	-	-
WCc05	B2	3.77	5.48	3.73	4.42	−45.36	1.06	−14.71
WCc10	B1	3,95	5.54	4.23	4.82	−40.25	−7.09	−18.05
WCSf0	B3	3.87	4.83	-	-	−24.81	-	-
WCSf05	B2	3.77	4.80	3.36	4.27	−27.32	10.88	−11.71
WCSf10	B1	3.75	5.38	3.90	5.06	−43.47	−4.00	−25.89
WCSc0	B3	3.17	4.99	-	-	−57.41	-	-
WCSc05	B2	3.30	4.91	3.16	4.46	−48.79	4.24	−26.01
WCSc10	B1	3.85	5.47	3.79	5.12	−42.08	1.56	−24.80

## Data Availability

The data presented in this study are available on request from the corresponding authors.

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
