# Peer review of "Cracking Behavior and Deflections in Recycled-Aggregate Beams Reinforced with Waste Fibers Subjected to Long-Term Constant Loading"

_materials, 2023, doi:10.3390/ma16103622_

Round 1
Reviewer 1 Report
The introduction section contains a literature review. It is necessary to single it out separately, designating it as a relative work. Then the relevance and purpose of the research will be more emphasized in the introduction. When compiling a literature review, it is not enough to simply enumerate and describe the calculation methods. It is necessary to reflect the author's attitude to the quoted sources. The experiments were carried out in accordance with the standards. It would be nice to point them out.
Selected remarks
Fig.6 has no dimension along the vertical axes in the text before figure 6 this information is also missing
Figure 15. Beam deflection diagrams do not indicate the dimensions along the y-axis. The text before Figure 15 also lacks this information. The visual accompaniment of Figure 14 would not be good enough to give quantitative comparisons and deviations.
Figure 8-15 does not indicate the measurement errors of the mathematical expectation, variance. By the way, this is not in the text either.
Figure 7 does not show the form of crack opening, as it is declared in the text after the caption to Figure 7.
In the materials and methods section, the technology of four-point bending is poorly described: recording equipment, distance between indeters and prisms, loads.
Conclusions add quantitative results and not words about improving compressive strength and elastic moduli.
Reviewer 2 Report
The topic is very interesting and the experimental stand is ingenious. There are some issues that need to be addressed before the manuscript could be accepted to publication.
Line 16 - the compressive strength and modulus of elasticity can not be tested but determined experimentally - please rephrase
Line 18 - "...stand which allows..."
Line 28 - please avoid personal statements such as "we" or "you". Rephrase using an impersonal form
Line 35 - what are those "some properties" the authors refer to?
Line 37-38 - the statement is incomplete. Please rephrase or complete the sentence
Line 41 - what do you mean by "temporary characteristics"? Are the authors referring to "short term
Lines 45-47 - the statement is ambiguous. Please rephrase.
Line 106 - what do you mean by "basic research"?
Line 113 - please substitute "the assumption" by "the parameters of..."
Lines 149-150 - why did you not test all 9 specimens for modulus of elasticity and compressive strength?
Line 152 - did you mean longitudinal reinforcing bars of 10 mm diameter instead of rods?
Line 155 - the concrete cover was 15 mm
Line 167 - beams were subjected to a four-point bending test
Figure 6 - how often were the temperature and air humidity measured?
Line 220-221 - the sentence is unfinished. Please rephrase or finish the sentence
Line 240 - Figure 7 presents the crack pattern for the beams made of Sc mix.
Figure 7 - "crack after 1000 days"
Figure 7 - did the authors consider only the vertical cracks? I would have expected, at least for the beams containing fibers, to see several smaller cracks. Was there a threshold for crack opening below which they were ignored?
Figures 8-15, caption - please substitute "calculating" by "analytical"
Line 341 - please check the decimal separator
Line 347 - "EC2 overestimates the results by ..."
As a general rule, the entire manuscript, with the exception of Conclusions section and when referring to tables and figures should be written in the past tense because it presents work that was done. Conclusions section should be written in present tense because it represents a scientific truth that is generally valid over time.
